# Doxycycline Induced Pancreatitis: An Uncommon Complication of a Common Drug

**DOI:** 10.3390/pharmacy10060144

**Published:** 2022-10-30

**Authors:** William Reiche, Faith Abodunrin, Chris Destache, Rajani Rangray, Manasa Velagapudi

**Affiliations:** 1Department of Medicine, CHI Health Creighton University Medical Center, Omaha, NE 68124, USA; 2Department of Pharmacy Practice, Creighton University School of Pharmacy and Health Professions, Omaha, NE 68178, USA; 3Division of Gastroenterology and Hepatology, CHI Creighton University Medical Center, Omaha, NE 68124, USA; 4Division of Infectious Diseases, CHI Creighton University Medical Center, Omaha, NE 68124, USA

**Keywords:** doxycycline, DIAP, drug induced acute pancreatitis, DIP, adverse drug reaction

## Abstract

We describe the risk factors for the development, timing, and severity of doxycycline induced acute pancreatitis (DIAP) in two case reports and a review of prior published cases, to better understand DIAP. Clinicians must maintain a high level of suspicion for DIAP in patients presenting with acute pancreatitis, while on doxycycline therapy. The latency and severity of DIAP are variable, making diagnosis challenging. Treatment includes bowel rest, hydration, and discontinuation of doxycycline.

## 1. Introduction

Drug-induced pancreatitis (DIP) is a rare entity, which accounts for 0.1 to 2% of all acute cases of pancreatitis (AP) [1]. DIP is usually a diagnosis of exclusion after more common etiologies are ruled out. Several medications, including mesalamine, azathioprine, simvastatin, and tetracycline antibiotics have been implicated in AP [2]. While most often, the clinical course of DIP is mild to moderate, severe cases of DIP can occur, making the description of DIP important [1]. The exact mechanism by which DIP occurs is unknown but may be due to the accumulation of toxic metabolites [3]. Doxycycline is a tetracycline antibiotic that has been infrequently associated with AP. The current literature is limited to a few case reports of doxycycline-induced acute pancreatitis (DIAP). Doxycycline use has increased in recent years, which might have increased the occurrence of this rare adverse effect. Hence, physicians should consider a diagnosis of DIP in patients who develop nausea, vomiting, and epigastric pain, while on doxycycline therapy. A careful history, with special attention to the onset of symptoms of presentation, in association with the start of doxycycline is key to the diagnosis.

In this case series, we report two cases of patients diagnosed with DIAP after careful exclusion of other causes of AP. We also performed a systematic review of the current literature regarding DIAP.

## 2. Case Description

### 2.1. Case 1 Presentation

A 60-year-old woman presented to the emergency department with a two-day history of severe nausea, vomiting, and epigastric pain. She denied diarrhea, constipation, hematochezia, or melena. Her past medical history was pertinent for Crohn’s disease, type 2 diabetes mellitus, hypothyroidism, chronic back pain with radiculopathy and hypertension. She had undergone hemicolectomy, ileostomy, and cholecystectomy in the past. She quit smoking three years prior and denied alcohol or recreational drug use.

Her outpatient medications included:Gabapentin,Hydrocodone-acetaminophen,Bupropion,Atorvastatin,Aspirin,Cyclobenzaprine,Dicyclomine,Esomeprazole,Fenofibrate,Glimepiride,Hydroxyzine,Levothyroxine,Metoprolol tartrate,Mirabegron,Quetiapine,Ondansetron,Furosemide,Famotidine,Celecoxib.VedolizumabCeftriaxoneDoxycycline

On arrival, the patient was afebrile, with a temperature of 36.3 C and hemodynamically stable with an initial blood pressure of 156/83 mmHg. Her heart rate was 85 beats/minute and respiratory rate 21 breaths/minute. A physical examination revealed she was oriented to person, place, and time and she had epigastric and right upper quadrant tenderness. The patient’s skin did not appear jaundiced and no scleral icterus was observed.

The patient’s laboratory results were as follows:Sodium 141 mmol/L (normal 135 to 145 mmol/L),Potassium 4.3 mmol/L (normal 3.7 to 5.1 mmol/L),Serum lipase 6699 µ/L (normal 73 to 393 µ/L),BUNCreatinine 1.94 mg/dL (normal range 0.60 to1.30 mg/dL),Blood glucose 174 mg/dL (normal 70 to100 mg/dL),Calcium 9.6 mg/dL (normal range 8.5 to 10.5 mg/dL),Her lipid panel were within normal limits.AST 15 µ/L (normal range 10 to 40 µ/L)ALT 20 µ/L (normal range 12 to 78 µ/L)ALP 98 µ/L (normal range 33 to 138 µ/L),Total bilirubin 0.2 mg/dL (normal range 0.0 to 1.5 mg/dL),TriglyceridesIgG subclasses were within normal limits.Leukocytes 9.2 k/µL (normal range 4 to 12 k/µL),Platelets 176 k/µL (normal range 140 to 440 k/µL),C-reactive protein 85 mg/dL (normal range < 9 mg/dL).

Computerized tomography (CT) of the abdomen and pelvis confirmed acute interstitial pancreatitis (Figure 1). No obvious necrosis or peripancreatic fluid was observed on imaging. Of note, a CT of the abdomen and pelvis, several weeks prior to this current presentation, revealed a normal-appearing pancreas. Lipase was also 85 µ/L and hemoglobin A1c 6.5% two weeks prior. No pleural effusions were noted on chest X-ray. Her Bedside Index of Severity in Acute Pancreatitis (BISAP) score was 2. No endoscopic retrograde cholangiography (ERCP) was performed prior to her having symptoms.

While vedolizumab has been associated with acute pancreatitis, we had a low suspicion for vedolizumab being the causative agent, as it was started two months prior and previous reports have suggested the onset of acute pancreatitis is rapid, following administration [4,5]. She was on IV ceftriaxone and oral doxycycline 100 mg twice daily for the treatment of septic arthritis two weeks prior. Ceftriaxone has been reported in several case reports to cause pancreatitis within two to seven days of medication initiation [6,7]. The patient was also taking furosemide (Class Ia), acetaminophen (Class II), and atorvastatin (Class III) which are known to cause acute pancreatitis; however, she had been taking these medications for several years prior to her developing acute pancreatitis [8]. Given her pancreatitis occurred outside of the reported latency period for ceftriaxone and vedolizumab, doxycycline was determined to be the most likely causative agent. With aggressive fluid resuscitation, pain management, and the discontinuation of doxycycline, her symptoms gradually improved. A repeat lipase five days after was 85 µ/L. A repeat CT scan of the abdomen and pelvis two weeks after revealed, necrotizing pancreatitis with walled off necrosis, measuring up to 15.4 cm.

### 2.2. Case 2 Presentation

A 91-year-old woman with a history of cholecystectomy, presented with two days of increased confusion, malaise, and severe generalized abdominal pain. She reported constipation; denied nausea, vomiting, hematochezia, melena, or trauma to the area. Four weeks prior to the admission, she underwent an open reduction and internal fixation of her left elbow fracture that was complicated by wound dehiscence. Wound cultures from her wound were positive for methicillin-resistant *Staphylococcus aureus* and *Enterobacter ludwigii*. She was treated with doxycycline 100 mg twice daily for 22 days. She never smoked, denied alcohol, or recreational drug use.

Her outpatient medications included:AcetaminophenBisacodylBumetanideDigoxinDocusate sodiumGabapentinIsosorbide dinitrateLevofloxacinMetoprolol tartratePantoprazoleRivaroxabanSenna-docusateSertralineSimvastatinTramadolDoxycycline

On arrival, she was afebrile, her heart rate was 105/min, hypotensive with a blood pressure of 80/32 mmHg, and tachypneic at 21 breaths/min. She was oriented to person, place, and time; however, she appeared ill. There was no scleral icterus, and her mucous membranes were dry. There was no jaundice and she had diffuse tenderness on the abdominal exam without peri umbilical or flank ecchymosis.

The patient’s laboratory results were as follows:BUN 56 mg/dL (6–24 mg/dL),Creatinine 2.35 mg/dL (0.60 to 1.30 mg/dL),ALP 223 µ/L (33 to 138 µ/L),Leukocyte count 26.7 k/µL (4.0 to 12.0 k/µL),Platelet count 156 k/µL (140 to 440 k/µL),Calcium 6.4 mg/dL (8.5 to 10.5 mg/dL),Lactic acid 2.8 mmol/l (0.4 to 2.0 mmol/l),Lipase 301 µ/L (73 to 393 µ/L).AST 40 µ/L (normal range 10 to 40 µ/L)ALT 12 µ/L (normal range 12 to 78 µ/L)Total bilirubin 1.1 mg/dL (normal range 0.0 to 1.5 mg/dL)Triglycerides and IgG subclasses were within normal limits.C-reactive protein 100 mg/dL (normal range < 9 mg/dL).

The CT abdomen was notable for extensive intra pancreatic and peripancreatic edema (Figure 2). A chest X-ray was obtained and was negative for pleural effusions. Her BISAP score was 5. Based on a thorough review of the patient’s chart, she did not have any obvious cause of AP. No endoscopic retrograde cholangiography (ERCP) was performed recently. She had been taking acetaminophen (Class II), sertraline (Class IV), and simvastatin (Class Ia), which are known to cause pancreatitis; however, she had been taking these medications for several years prior to the development of her pancreatitis, making these medications less likely causes [8]. Given her recent doxycycline use and no other identifiable cause, DIAP was diagnosed. She was transferred to the intensive care unit for hypovolemic shock. She required fluid resuscitation with vasopressors and the discontinuation of doxycycline which resulted in a full recovery.

## 3. Discussion

AP is most often caused by gallstone disease or alcohol. Other clinically relevant etiologies include hypertriglyceridemia, hypercalcemia, trauma, post-ERCP autoimmune, and infections. Acute pancreatitis can also be iatrogenic following endoscopic retrograde cholangiography (ERCP). When common causes are ruled out, clinicians often default to a diagnosis of idiopathic pancreatitis, which occurs roughly 10% of the time. However, the adverse effects of medications should be included in the differential for idiopathic pancreatitis when other causes are absent. To date, more than 213 medications have been implicated in causing AP [9]. Drug-induced pancreatitis (DIP) is a rare but potential etiology of acute pancreatitis and should be considered, especially in at-risk populations. Elderly patients, women, patients with an advanced HIV infection, or patients with type 2 diabetes mellitus, hyperlipidemia, or inflammatory bowel disease, have been noted to be particularly at risk of DIP [10]. It can be challenging to determine which medication is the cause of pancreatitis, as patients may be taking multiple medications known to cause pancreatitis. Importantly, the time to onset of symptoms with relation to starting the medication may aid in identifying the causative agent. However, causality becomes difficult as it can be difficult, ethically, to perform a rechallenge. 

DIP is usually classified as class I to IV, depending on the existing supporting evidence in the literature [8]. Class I is defined as having at least one case report that has determined the drug caused acute pancreatitis after rechallenging. Class II is defined as having a consistent lag period between when the drug exposure occurred and when pancreatitis developed, with at least four cases reported in the literature. Up to 75% of the time, most cases reported are classified as Class II. Class III drugs have two or more case reports written; however, a rechallenge or lag period has not been determined. For Class IV drugs, a rechallenge or lag period has not been elucidated; and less than two case reports have been reported. Doxycycline has been assigned a Class Ic designation, as there is at least one case report in humans without a positive rechallenge [9]. Additional classification of DIAP using the Naranjo scale, has been limited by the lack of consistent reporting in prior case reports (Table 1). 

Systematic data on DIAP are lacking, as a result, our current understanding of DIAP is based entirely on case reports. A systematic literature review was performed with the help of a medical librarian, using the keywords “doxycycline” and “acute pancreatitis”. All previous case reports mentioning the above keywords were considered. Latency refers to the time from the initiation of the drug to symptom onset. Based on the case reports, DIAP has a variable latency. Prior case reports describe the onset occurring as early as one day after the initiation of doxycycline and occurring up to 14 days after starting the medication. In our first case, the symptoms occurred 14 days after doxycycline initiation, compared to 22 days in the second case. 

The standard dosing for doxycycline is 100 mg, twice daily; however, three of the included case reports have daily doses of more than 200 mg. There did not appear to be an impact on the severity of DIAP, given these higher doses as none of the patients required ICU level care. 

Consistent with the prior reported risk factors for developing DIP, of the 11 available reported cases of DIAP, eight (73%) occurred in women. While severe AP was noted in several of the eldest patients in the cases, younger patients (ages 51 and 52) also had severe AP. The reason elderly patients are thought to be at an increased risk of DIAP is the significant amount of polypharmacy which can be present. Our two cases exhibit this nicely, as our patient in case one was taking 22 medications and our patient in case two was taking 16 medications. Our first case had a history of type 2 diabetes mellitus and inflammatory bowel disease, two of the most common chronic diseases associated with the development of DIAP [9]. Patients with inflammatory bowel disease are thought to have an increased risk of not only DIAP but may be more likely to develop AP, given the pro-inflammatory state of their disease [2].

The severity of DIAP has been reported as variable in the past, four prior case reports have noted severe pancreatitis resulting in intensive care unit (ICU) monitoring while, five cases have not required ICU level care [16]. Our two included cases highlight the variable severity of DIAP. In case two, the BISAP score was 5 and careful monitoring in the ICU was required. While the patient in case one improved with intravenous fluid resuscitation in the hospital.

We used the Naranjo scale to determine the odds of doxycycline being the cause of the acute pancreatitis. More than 9 was considered a definite adverse drug reaction (ADR), 5 to 8 probable ADR, 1 to 4 possible ADR, and 0 doubtful ADR. The Naranjo scale for both cases was 3, given the known previous conclusive reports on the reaction with doxycycline (+1), the adverse event appearing after doxycycline was given (+2), the adverse reaction improvement after the drug was discontinued (+1), and the alternative causes that could, on their own, have caused the reaction (−1). However, given the extended duration of the alternative medications without reaction and the reaction occurring shortly after doxycycline initiation, the Naranjo scale downgrades the severity scoring of the adverse reaction. Additionally, due to the severity of the adverse reaction, rechallenge was not attempted in our cases and has not been attempted in review of previous case reports. This can be problematic as cause and effect are generally established on the basis of a rechallenge test. None of the other case reports utilized the Naranjo scale. Additionally, based on the outpatient medications, we could not link AP to any of their outpatient medications. Ceftriaxone was started at the same time as doxycycline in case one, making the differentiation of these two medications difficult; however, pancreatitis started outside of the noted latency period for ceftriaxone. Thus, it is possible for doxycycline to be the cause of AP in both cases, given the time interval and stopping the medicine allowed both patients to resolve their acute pancreatitis episode.

The treatment for DIAP is the discontinuation of doxycycline therapy and aggressive IV fluid resuscitation. Quite often, there may be multiple medications the patient is taking which may cause acute pancreatitis. In this instance, it may be best to discontinue the potential causative agents and, in a step-wise manner, reintroduce the medications after a thorough risk versus benefit discussion has occurred. It can be difficult to completely ensure doxycycline caused AP, as a cause of pancreatitis may not be found in up to 10% of the time. If antibiotic coverage is still warranted for the patient, an alternative antibiotic can be easily selected as often times susceptibilities are not limited to doxycycline alone. The exact mechanism of DIAP is not known. Some postulates for drug-induced pancreatitis are direct cellular damage, immune-medicated inflammation, and metabolic effects [19]. It is possible that toxic metabolites are released during the biotransformation of doxycycline, which affects the pancreas. There is room for further research in determining the pathogenesis of DIAP.

## 4. Conclusions

Clinicians must maintain a high level of suspicion for DIAP in patients presenting with AP symptoms and risk factors for DIP, in the absence of the classical causes of AP. The latency and severity of DIAP are variable, making diagnosis challenging. It is not clear if gender predilection for DIAP exists. Females are more likely to develop DIAP, based on the cases reported so far. Age is not an independent risk factor for DIAP as it affects patients of all ages. DIAP needs additional consideration as doxycycline prescriptions have increased in recent years. The BISAP score is a useful tool in determining the prognosis of DIAP and physicians should incorporate this when providing care. DIAP is primarily treated by discontinuing doxycycline in addition to measures typically used in acute pancreatitis. It is reasonable to prevent further exposure to doxycycline and the medication can be added to the patient’s allergy list.

## Figures and Tables

**Figure 1 pharmacy-10-00144-f001:**
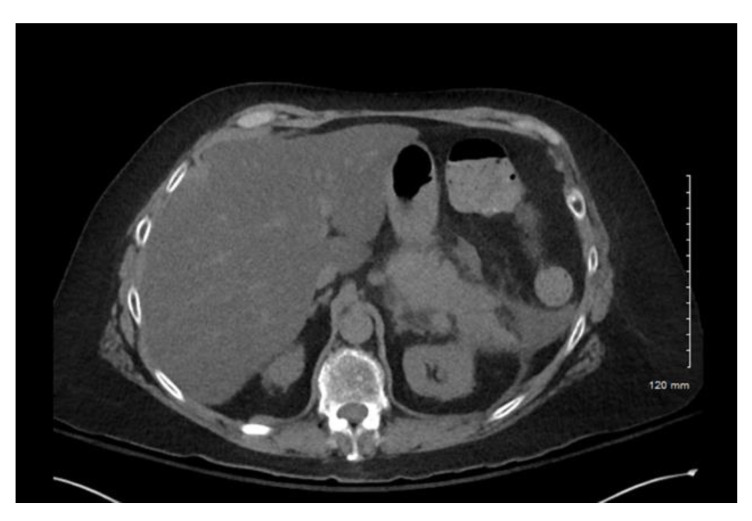
Case 1 CT abdomen and pelvis. Acute interstitial pancreatitis.

**Figure 2 pharmacy-10-00144-f002:**
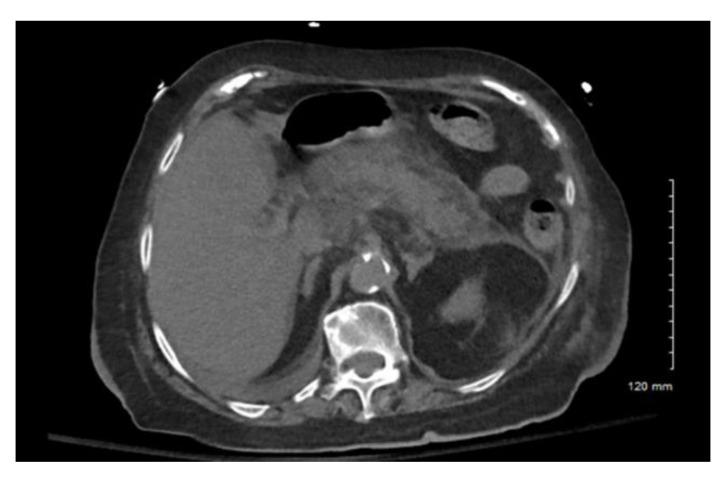
Case 2 CT abdomen & pelvis. Severe peripancreatic and pancreatic edema.

**Table 1 pharmacy-10-00144-t001:** Previous case reports on DIAP.

Ref.	Age	Gender	Doxycycline Dose (mg/d)	Latency (Days)	ICU	BISAP Score	Other Potential Meds
Inayat et al. [11]	58	F	400	2	N		
Rawla et al. [12]	52	F	200	7	Y		gabapentin oxycodone
Wachira et al. [13]	21	F		15	N		
Ocal et al. [14]	33	F	1000	3	N		ornidazole
Achecar Justo et al. [15]	75	F	400	14	N		
Moy et al. [16]	51	M	200	3	Y	3	
Paulraj et al. [3]	55	M	200	7	N		fenofibrate marijuana
Shah et al. [17]	76	F	200	7	Y	4	sertraline
Eland et al. [18]	23	M	200	1			
Case 1	60	F	200	14	N	2	ceftriaxone
Case 2	91	F	200	22	Y	5	

F, female; M, male; ICU, intensive care unit; Y, yes; N, no; BISAP, Bedside Index for Severity in Acute Pancreatitis.

## Data Availability

Not applicable.

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
