# Peer review of "Doxycycline Induced Pancreatitis: An Uncommon Complication of a Common Drug"

_pharmacy, 2022, doi:10.3390/pharmacy10060144_

Round 1

Reviewer 1 Report

The introduction, methods and results are very impressively clear and well explained. Case studies highlighting drug induced AP are significant and there is a need to publish this data. 

The main question being addressed by the manuscript is whether doxycycline can be a causative agent to cause pancreatitis. This is an interesting case study because although a rare adverse event, doxycycline can be known to induce AP in patients. Case studies such as the one in the manuscript also highlight other causative agents and is able to eliminate them by reasoning

It is not entirely an absolutely novel case however, such cases need to be documented to publish awareness and support other HCP in problem solving their cases. This case study is novel as other supposed pancreatitis causing agents were present in the patients medication list

It is very thoroughly written and reviewed. It follows a clear scientific train of thought.

The authors of the study present the cases and support their conclusions with data presented. 

Author Response

THe authors thank this reviewer for their comments regarding these case reports. 

Reviewer 2 Report

1) The description of the two cases of doxycycline-induced acute pancreatitis, and the review of the reported cases in the literature are of clinical interest.

2) The dose, duration of treatment and chronological sequence between drug administration and the onset of pancreatitis symptoms should be indicated regarding the other potential drug-induced pancreatitis (in case 1 gabapentin, acetaminophen, atorvastatin, furosemide, vedolizumab, and ceftriaxone; and in case 2 acetaminophen, gabapentin, sertraline and simvastatin)

3) Table 1 should be headed with a title. For example, reported cases of doxycycline-induced acute pancreatitis. It is suggested to include information on the duration of the treatment in the column of the daily dose of doxycycline. It is also suggested to include a column with the outcome of acute pancreatitis (full recovery, sequelae or fatal outcome).

4) In the discussion, the biological plausibility or mechanism of action suggested regarding doxycycline-induced pancreatitis should be discussed.

5) In the cases reported in the literature, it should be described if causality algorithms were used, such as the Naranjo scale or others, and the category or score of the scale.

Author Response

  1. The authors thank this reviewer for the comment.  
  2. the length of therapy for the concomitant drugs is "several years"
  3. The authors have included a title for Table 1. 
  4. The authors have added a hypothesized mechanism for this DDI: 

    The exact mechanism of DIAP is not known. Some postulates for drug-induced pancreatitis are direct cellular damage, immune-medicated inflammation, and metabolic effects [19]. It is possible that toxic metabolites are released during the biotransformation of doxycycline which affects the pancreas. There is room for further research in determining the pathogenesis of DIAP.

  5. none of the other case reports utilized the Naranjo scale.  this is discussed in the Discussion section. 

Reviewer 3 Report

This report includes important cases of doxycycline-induced pancreatitis.

This study explores whether doxycycline can be linked to pancreatitis. This study adds important cases to the literature of drug induced AP. 

The paper is well-written with sound methodology in addition to robust results. The conclusion precisely responds to the answers the authors have proposed in the manuscript, and the results are largely consistent with the evidence.

 I only have one minor comment. Please check the #2 reference.

Author Response

all the references were re-examined to ensure documentation accuracy.